# Association between Tumor Mutational Burden, Stromal CD8^+^ Tumor-Infiltrating Lymphocytes, and Clinical Factors in Cervical Cancers Treated with Radiotherapy

**DOI:** 10.3390/cancers15041210

**Published:** 2023-02-14

**Authors:** Hanguang Ruan, Takahiro Oike, Hiro Sato, Ken Ando, Tatsuya Ohno

**Affiliations:** 1Department of Radiation Oncology, Gunma University Graduate School of Medicine, 3-39-22, Showa-machi, Maebashi 371-8511, Gunma, Japan; 2Gunma University Heavy Ion Medical Center, 3-39-22, Showa-machi, Maebashi 371-8511, Gunma, Japan

**Keywords:** cervical cancer, radiotherapy, immune response, tumor mutational burden, CD8^+^ tumor-infiltrating lymphocyte, ARID1A

## Abstract

**Simple Summary:**

We retrospectively analyzed the association between tumor mutational burden (TMB) and/or the density of stromal CD8-positive tumor-infiltrating lymphocytes (CD8^+^TILs) and clinical factors in 44 patients with squamous cell carcinoma of the uterine cervix treated with definitive radiotherapy. We found that (i) TMB was not associated with CD8^+^TIL density; (ii) TMB-high plus CD8^+^TIL density-low status predicted a worse prognosis than either of the two factors alone; and (iii) TMB-high or CD8^+^TIL density-high status was associated with the presence of *ARID1A* mutations. These data suggest no association between TMB and CD8^+^TIL density, but involvement of *ARID1A* mutations, in antitumor immune responses in patients with cervical cancers treated with radiotherapy.

**Abstract:**

Background: Tumor mutational burden (TMB) and stromal CD8-positive tumor-infiltrating lymphocytes (CD8^+^TILs) serve important roles in antitumor immune responses to radiotherapy. This study aimed to elucidate the association between TMB, CD8^+^TILs, and clinical factors in patients with cervical cancer treated with radiotherapy. Methods: Patients with squamous cell carcinoma of the uterine cervix treated with definitive radiotherapy, and with available somatic mutation data and immunohistochemical staining data from identical tumor tissues, were enrolled retrospectively. The association between TMB and/or CD8^+^TIL density and patient characteristics, mutation profiles, and treatment outcome was analyzed. Results: The study analyzed 44 patients (median follow-up period, 61 months). There was no significant correlation between TMB and CD8^+^TIL density, or between TMB or CD8^+^TIL density and patient characteristics. TMB-high or CD8^+^TIL density-low status was associated with worse overall survival and distant metastasis-free survival; the predictive value of these factors became greater when used in combination. TMB-high or CD8^+^TIL density-high status was associated with *ARID1A* mutations. Conclusions: These data indicate independence of TMB and CD8^+^TIL density and the involvement of ARID1A alterations in antitumor immune responses in patients with cervical cancers treated with radiotherapy, warranting further mechanistic research and prospective validation.

## 1. Introduction

Cervical cancer is the fourth leading cause of death among all cancers worldwide [1]. At the early stage, cervical cancer is, in large part, manageable by implementing primary and secondary preventions (i.e., vaccination and screening) [2]. Accumulating evidence suggests that the adoption of human papillomavirus (HPV) vaccines contributes to a significant decrease in HPV infection, HPV lesions, and cervical cancer occurrence. Screening using Pap smear also contributes to a significant decrease in cervical cancer mortality. However, discomfort and embarrassment regarding the vaccination and Pap smear testing prevent these measures from being widespread, leading to the not-insignificant incidence of locally advanced diseases. Radiotherapy plays the role as a definitive treatment for locally advanced cervical cancer [3]. Recent advancement in image-guided techniques, especially those on brachytherapy, has improved the treatment outcome of this disease dramatically [4]. However, some patients still suffer from recurrence from the irradiated site or from distant metastasis after the radiotherapy. This highlights the necessity for obtaining deeper understanding of the antitumor effects of radiotherapy for cervical cancer, upon which better multi-disciplinary treatment should be developed.

Various combinations of chemotherapy, targeted therapy, and radiotherapy have been explored for the management of locally advanced cervical cancer [5]. Among them, immune checkpoint blockade has attracted broad interest, and a number of clinical trials to test its efficacy are ongoing. Accumulating evidence highlights the important roles of immune responses in the antitumor effects of radiotherapy [6]. Irradiated tumor cells release neo-antigens which are recognized by antigen-presenting cells; this activates antitumor immune responses mediated by T cells [7,8,9]. Tumor mutational burden (TMB, with a unit of mut/Mb), is the index calculated by dividing the number of somatic mutations by the sequenced lengths of the genome [10,11,12]. TMB is considered to be associated with neo-antigen levels because neo-antigens are produced from somatic mutations of cancer cells [10,11,12]. There is robust evidence that TMB-high tumors respond to immune checkpoint inhibitors (ICI) [12,13]. Meanwhile, the impact of TMB status on radiotherapy outcome is unclear. Recently, Ota et al. and Yuan et al. reported a correlation between high TMB status and worse overall survival (OS) of patients with cervical cancer (*n* = 98) or esophageal cancer (*n* = 18) who received definitive radiotherapy, respectively [14,15], although the underlying mechanisms are unknown.

Stromal CD8-positive tumor-infiltrating lymphocytes (CD8^+^TILs) play pivotal roles in immune responses by eliminating cancer cells [16,17]. Several studies have demonstrated a correlation between high CD8^+^TIL status and a favorable outcome after radiotherapy for cervical cancer [15,16,17,18,19]. Considering the above-mentioned flow of post-irradiation antitumor immune responses (from the release of neo-antigens to T cell mediated elimination of tumor cells), it seems reasonable that high TMB tumors are enriched with CD8^+^TILs. However, the association between TMB and CD8^+^TILs has not been elucidated in clinical samples. In addition, the association between TMB and/or CD8^+^TILs with patient characteristics, mutation profiles, and treatment outcomes remains unclear. Thus, we performed a study aiming to elucidate the association between TMB, CD8^+^TILs, and clinical factors in patients with cervical cancer treated with definitive radiotherapy by collecting the cases available for somatic mutation data and immunohistochemical staining data from identical tumor tissues.

## 2. Materials and Methods

### 2.1. Study Cohort

The patients who met the following criteria were retrospectively enrolled to this study. Inclusion criteria: (i) newly diagnosed and untreated uterine cervical cancer; (ii) pathologically diagnosed as squamous cell carcinoma; (iii) the International Federation of Gynecology and Obstetrics (FIGO) 2009 stage IB–IVA; (iv) treated with definitive radiotherapy at Gunma University Hospital during 2009 to 2013; (v) available for the data on age, gynecological examination, imaging, and treatment contents; (vi) available for somatic mutation data from tumor biopsy specimens obtained before initiation of radiotherapy (pre-RT); and (vii) available for immunohistochemical staining data for tumor tissues obtained at the pre-RT and at 10 Gy (inter-RT) timepoints. Exclusion criteria: patients for whom sample acquisition may lead to deterioration of their general condition.

### 2.2. Radiotherapy

Radiotherapy was performed as previously described [20]. Briefly, radiotherapy was performed by combining external beam radiotherapy (EBRT) with brachytherapy. For EBRT, pelvic irradiation was delivered for 50 Gy in 25 fractions; five fractions per week. A central shielding technique was employed for the last 30 Gy for the cases with stage IB–II tumors (≤40 mm) without lymph node involvement, and for the last 20 Gy for the other cases. Boost irradiation for the positive lymph nodes was performed for 6–10 Gy/3–5 fractions. Concomitant with EBRT, weekly cisplatin (40 mg/m^2^) was given for the cases with stage III–IVA disease, a bulky tumor (i.e., >40 mm), or lymph node involvement. For brachytherapy, image-guided planning based on computed tomography (CT) was employed based on the recommendations from the Groupe Européen de Curiethérapie and the European Society for Radiotherapy and Oncology, in which a total of 24 Gy was delivered in four fractions, one fraction per week, using a high dose rate ^192^Ir source. Fletcher-Suit Asian Pacific applicators, in combination with trocar point needles, were used.

In this study, the day of initiation of radiotherapy was noted as Day 1. Post-treatment surveillance was performed every 1–3 months for the first 2 years and every 3–6 months thereafter. Disease status was evaluated by combining gynecological examination and CT or magnetic resonance. OS, pelvic recurrence-free survival (PRFS: defined as no evidence of recurrence in the pelvic region), and distant metastasis-free survival (DMFS) were recorded.

### 2.3. Assessment of TMB

TMB was assessed as described previously [14]. Briefly, the formalin-fixed paraffin-embedded (FFPE) tissue was prepared from tumor biopsy samples, from which DNA was extracted using a QIAamp DNA FFPE tissue kit (QIAGEN, Hilden, Germany). After examination of DNA fragmentation by using a TaqMan RNase P detection reagent kit (Thermo Fisher Scientific, Waltham, MA, USA), amplicon libraries were prepared by using an Ion AmpliSeq library kit 2.0 (Thermo). Sequencing was conducted by using the Ion Torrent and Ion AmpliSeq Comprehensive Cancer Panel (Thermo). The cancer panel contains >95% of the exon sequence of 409 cancer-related genes. Genome Reference Consortium Human Build 37 (hg19) was used as the reference for the sequence. Single nucleotide polymorphisms were determined with those for the subject NA12878 registered in the 1000 Genome project as a reference.

Somatic mutations were determined using the following criteria [21]: total coverage >20; variant coverage >10; variant frequency >15%; minor allele frequency <0.1%; SIFT value 0–0.5; Polyphen-2 value, 0.05–1.0; and Grantham value 25–215. The dbSNP was referenced to distinguish the called variants from single nucleotide polymorphisms. Samples with read-on-target <25, depth <100, and total number of mutations/Mb >100 were excluded. TMB value was calculated by dividing the number of somatic mutations for individual cases by the targeted sequence length (i.e., 1.688650 Mb).

### 2.4. Human Papillomavirus Genotyping

HPV genotyping was performed on DNA samples prepared in Section 2.3. using the PapiPlex PCR method (GeneticLab, Sapporo, Japan) that targets genotypes 6, 11, 16, 18, 30, 31, 33, 35, 39, 45, 51, 52, 56, 58, 59, and 66 [21].

### 2.5. Immunohistochemical Analysis of CD8^+^TILs

Immunohistochemical staining of CD8 was performed on FFPE tumor specimens obtained at the pre-RT or inter-RT timepoints, as described previously [16]. Briefly, 4 μm thick paraffin sections were de-waxed with xylene at room temperature and then re-hydrated through a series of graded ethanol solutions. Blockade of endogenous peroxidase activity was performed by the addition of 0.3% hydrogen peroxide for 10 min at room temperature. Sections were heated for 10 min at 121 °C in 0.01 mol/L citric acid (pH 6.0) for antigen retrieval. After blocking for 20 min at room temperature with 10% rabbit normal serum, sections were incubated overnight at 4 °C with a monoclonal anti-CD8 antibody (clone C8/144B, Agilent Technologies, Inc., Santa Clara, CA, USA; 1:800). Reaction with biotin-labeled secondary antibodies was performed for 20 min at room temperature, which was followed by reaction with peroxidase-labeled streptavidin for 20 min at room temperature. Finally, sections were incubated with diaminobenzidine for 5 min at room temperature to develop color.

Staining images were obtained under a BZ-9000 light microscope (Keyence Corporation, Osaka, Japan). CD8-positive cells were counted automatically using a BZ-X analyzer JP ver.1.4.1 (Keyence). CD8^+^TIL density was defined as the percentage of CD8-positive cells within the total population of nucleated cells in the stromal tissue [16,22]. To calculate CD8^+^TIL density, the cell count was performed in the hotspot area defined as the stromal tissue that contains the highest density of nucleated cells, where the tumor cells was excluded by visual inspection [16,22].

### 2.6. Statistical Analysis

Differences between two groups were assessed by the Mann–Whitney U test. Correlations between two groups were assessed using Spearman’s rank correlation test. Association of two categorical variables was assessed using Fisher’s exact test. The survival probability (i.e., OS, PRFS, and DMFS) was analyzed by using the Kaplan–Meier method and log-rank test. All statistical analyses were conducted using by GraphPad Prism 9 (GraphPad, San Diego, CA, USA) with statistical significance set at *p* = 0.05.

## 3. Results

This study analyzed 44 patients. The median follow-up period was 61 months (Table 1). The median value for TMB was 9.5 mut/Mb (interquartile range [IQR], 7.1–12.2 mut/Mb) (Figure 1a). The median value for pre-RT CD8^+^TIL density was 22.5% (IQR, 6.5–43.2%) (Figure 1b). The study cohort was analyzed hereafter by dichotomization with the median TMB or pre-RT CD8^+^TIL density.

The association between TMB or pre-RT CD8^+^TIL density and patient characteristics was analyzed. There was no association between TMB or pre-RT CD8^+^TIL density, and age, FIGO stage, tumor diameter, nodal status, HPV genotype, or concomitant use of chemotherapy (Table 2).

The correlation between TMB and CD8^+^TIL density was analyzed. Contrary to our expectation, there was no significant correlation between TMB and pre-RT CD8^+^TIL density, inter-RT CD8^+^TIL density, or the change in CD8^+^TIL density between the two timepoints (Figure 2a–c).

The 5-year OS rate for stage IB + II and stage III + IVA patients was 90.5% and 64.3%, respectively (Appendix A); this outcome is in line with that reported in previous studies [23,24,25]. The influence of TMB and/or pre-RT CD8^+^TIL density on treatment outcome was also examined. For TMB alone, there was no significant difference in OS, PRFS, or DMFS between TMB-high and -low patients, although there was a modest trend toward a worse OS or DMSF for TMB-high patients (Figure 3a–c). For pre-RT CD8^+^TIL density alone, patients with low CD8^+^TIL density showed significantly worse OS (*p* = 0.0031), whereas a trend toward a worse PRFS or DMFS was observed for patients with low CD8^+^TIL density (Figure 3d–f). Interestingly, the ability of the combination of TMB plus pre-RT CD8^+^TIL density to predict worse OS or DMSF was greater than that by either of the two factors alone, underscoring a strikingly poor OS or DMFS for patients with the TMB-high/CD8^+^TIL-low status (Figure 3g–i).

Recurrent mutations were observed in *PIK3CA* (43.1%), *ARID1A* (34.0%), *NOTCH1* (27.2%), *FBXW7* (25.0%), *FGFR3* (20.4%), *EP300* (18.1%), *FGFR4* (13.6%), and *TP53* (6.8%); this mutation spectrum was broadly consistent with that reported in the landmark studies [26,27]. The association between TMB or pre-RT CD8^+^TIL density and mutation profile was analyzed to identify genes whose mutation prevalence was >20%. The results showed that the TMB in *ARID1A*-mutant tumors was significantly greater than in *ARID1A*-wild-type tumors (*p* = 0.0008), whereas the TMB did not differ significantly between mutant and wild-type tumors with respect to *PIK3CA*, *NOTCH1*, or *FBXW7* (Figure 4a). A trend toward greater pre-RT CD8^+^TIL density in *ARID1A*-mutant tumors was observed (*p* = 0.11), whereas pre-RT CD8^+^TIL density did not differ significantly between mutant and wild-type tumors with respect to *PIK3CA*, *NOTCH1*, or *FBXW7* (Figure 4b).

## 4. Discussion

The present study found that TMB was not associated with CD8^+^TIL density in patients with cervical cancer treated with definitive radiotherapy. A correlation between TMB and CD8^+^TIL density was absent at both the pre-RT and inter-RT timepoints. Supporting this notion, an increase in the predictive ability of the combination of TMB plus pre-RT CD8^+^TIL density above that of each individual factor alone suggests that both are independent of each other. These results are in contrast to our expectation from the perspective that a greater number of neo-antigens will lead to stronger T cell-mediated antitumor immune responses, suggesting the existence of an as-yet-to-be elucidated mechanistic link between them. We expected a positive correlation between TMB and CD8^+^TILs because TMB is known to correlate with neoantigen load [28,29,30], and another analysis of The Cancer Genome Atlas reported that the group containing cervical squamous cell carcinoma and endocervical adenocarcinoma samples showed a positive correlation between neoantigen load and CD8^+^TIL numbers [31]. This discrepancy may be explained by the different patient backgrounds (e.g., endocervical adenocarcinoma was also included in the database analysis). Meanwhile, there have also been reports of malignancies, including breast and prostate cancer, in which neoantigen load does not correlate with CD8^+^TILs [32]. In future studies, the underlying mechanisms should be prospectively elucidated with a specific type of malignancy. Given the above, and as shown in our previous study [16], the present data suggest that it would be reasonable to analyze CD8^+^TIL density collectively rather than TMB alone as a prognostic marker for radiotherapy for cervical squamous cell carcinoma.

The present study is the first to report an association between *ARID1A* mutations and high TMB status in cervical cancer. Recent studies reported an association between *ARID1A* mutations and high TMB status in various types of cancer, including colorectal cancer [33,34,35], ovarian clear cell carcinoma (OCCC) [36], gastroesophageal cancer, non-small cell lung cancer, and endometrial cancer [34]. Tokunaga et al. investigated 7978 colorectal cancer cases and found that *ARID1A* mutant tumors had more genomically unstable natures (i.e., high TMB and high microsatellite instability [MSI] status) [33]. Okamura et al. investigated 301 colorectal cancer cases and found that TMB-high tumors were more prevalent in *ARID1A*-mutant cases than in -wild type cases (41% versus 3.1%; *p* < 0.001); also, MSI-high tumors were more prevalent in *ARID1A* mutant cases than in wild type cases (32% versus 2.9%; *p* < 0.001) [34]. Kamori et al. investigated the data at their institution and those from The Cancer Genome Atlas and found that TMB was higher in *ARID1A* mutant colorectal cancers than wild type counterparts in both datasets [35]. Kuroda et al. investigated 41 cases of OCCC and found that TMB was higher in *ARID1A* mutant cases than in wild type cases [36]. Interestingly, the study by Tokunaga et al. indicates that the association between *ARID1A* mutation and high TMB status seems to be true even in the absence of mutations involved with MSI [33]. This notion is consistent with the results of the present study where the mutations in *MLH1*, *MSH2*, and *MSH6* were not detected. The mechanisms by which *ARID1A* mutations contribute to high TMB status are yet unelucidated. ARID1A is a subunit of the SWI/SNF complex that functions as ATPase chromatin remodeler [37]. The SWI/SNF complex remodels the DNA-histone contacts that changes the accessibility of specific parts of the genome; this allows for the biological processes including transcription and DNA damage response. The SWI/SNF complex is considered to be a tumor-suppressor, and notably, the genes encoding the subunits of the SWI/SNF complex, in total, are found to be mutated in approximately 20% of call malignant tumors, rivaling the mutation prevalence for another famous tumor suppressor *TP53*. Evidence suggests that the SWI/SNF complex is recruited directly to the damaged DNA sites and promotes the repair of the damaged DNAs by remodeling the nucleosomes. From this standpoint, it is considered reasonable that the loss of function mutations in *ARID1A* contribute to high TMB status in the tumors. However, previous studies indicate that the mutations in other SWI/SNF genes (e.g., *PBRM1*, *SMARCA4*, and *SMARCB1*) are not always associated with high TMB status, warranting further research to elucidate the role of *ARID1A* mutation in high TMB [38]. We also found a trend toward greater CD8+TIL density in *ARID1A* mutant tumors. This result is in line with those reported by previous studies showing a higher expression of CD8 in ARID1A-mutant tumor tissues in patients with OCCC [34], and enrichment of CD8+ T cells, CD4+ T cells, and NK cells in ARID1A-mutant tumor tissues in patients with gastric cancer [39], although the underlying mechanisms are unknown.

An interesting sub-finding of the present study is that TMB-high plus CD8^+^TIL density-low status predicts a worse prognosis for OS and DMFS, but not for PRFS. These data indicate that this immunological status is associated with greater metastatic potential of locally advanced cervical cancers treated with definitive radiotherapy, although the underlying mechanisms are unknown. Thus, for patients who have the combined status, the intensity of systemic treatment may be increased to prevent metastasis. Multiple clinical trials are ongoing to test the combination of radiotherapy and ICIs (e.g., Pembrolizumab, Nivolumab, and Ipilimumab) [40]. PARP inhibitors may be another candidate considering their mechanism of action, since radiotherapy induces DNA double-strand breaks which are in part repaired by homologous recombination, a synthetic lethality partner with PARP [41]. These potential strategies await clinical validation.

The strengths of this study include the following. First, this study cohort was available simultaneously for pre-treatment FFPE tumor specimens, pre-treatment frozen tumor tissues, and inter-treatment frozen tumor tissues. Contrary to the patients treated with surgical resection, in general, it is highly difficult to obtain such set of tumor tissue samples from patients treated with radiotherapy. From this standpoint, we believe that the dataset obtained from this study cohort is of value. Secondly, the mutational analysis performed in this study was compatible with the clinical practice of precision medicine, where panel-based sequencing is utilized. The results of the present study support the notion that TMB is assessable by using the FFPE tumor specimens obtained from routine clinical practice. Thirdly, the patients analyzed in this study received standardized treatment and follow-up for sufficient periods (i.e., the median follow-up period of 61 months).

The limitations include the following. First, this study has a retrospective design; thus, the evidence level is not high. Secondly, this study cohort consists of a small number of participants. For this reason, multivariate analysis was not performed due to a small number of events. Thirdly, we were not able to perform immunohistochemical analysis for the molecules other than CD8^+^TILs due to insufficient tumor specimens. In addition to CD8 positive cells, cervical cancer TILs include CD4 positive cells, αβ T cells, γδ T cells, B cells, natural killer cells, and regulatory T cells (Tregs) [42]. In general, CD8- and CD4-positive cells are the predominant types for TILs, and, therefore, the enrichment of these cells is considered as a positive prognostic indicator for better outcome. Nevertheless, evidence suggests that not only the amount of a single immune cell fraction (e.g., CD8^+^TIL density analyzed in this study) but also the balance of their fractions (e.g., the CD4/CD8 ratio or the CD8/Treg ratio) are important in antitumor immune activity. In addition, the status for immune checkpoint molecules can also influence the treatment outcome. For example, the cases with high PD-1 expression on the CD8^+^TILs are associated with earlier post-treatment recurrence [43]. Furthermore, in light of clinical implementation, the markers used in daily clinical practice, such as Ki-67, could be investigated in combination with immune-associated molecules to pursue better predictive ability by the combined expression status.

## 5. Conclusions

Radiotherapy facilitates the release of neo-antigens from tumor cells; thus, the antitumor effect of radiotherapy can be, in part, exerted through antitumor immune responses. TMB is associated with the amount of neo-antigen in tumor cells, whereas CD8^+^TILs play a major role in eliminating tumor cells. In this study, in order to obtain a deeper understanding of the role of antitumor immune responses on the outcome of definitive radiotherapy, we focused on the association between TMB and CD8^+^TILs, which had not been directly analyzed in clinical samples. We retrospectively analyzed 44 patients with cervical squamous cell carcinoma treated with definitive radiotherapy, for whom pre-RT FFPE tumor tissues, and pre- and inter-RT frozen tumor samples were simultaneously available. TMB and CD8^+^TIL density was assessed by panel-based clinical sequencing and immunohistochemistry, respectively. As a result, there was no significant correlation between TMB and CD8^+^TIL density, or between TMB or CD8^+^TIL density and patient characteristics. TMB-high or CD8^+^TIL density-low status was associated with worse OS and DMFS. Interestingly, the predictive value of these factors became greater when used in combination. TMB-high or CD8^+^TIL density-high status was associated with *ARID1A* mutations. These data indicate the independence of TMB and CD8^+^TIL density and the involvement of ARID1A alterations in antitumor immune responses in patients with cervical cancers treated with radiotherapy, warranting further mechanistic research and prospective validation.

## Figures and Tables

**Figure 1 cancers-15-01210-f001:**
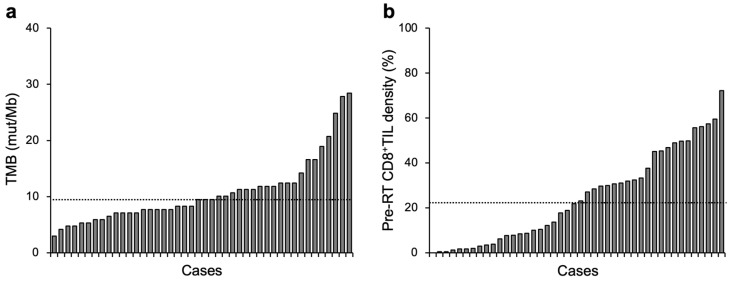
Overview of TMB (**a**) or pre-RT CD8^+^TIL density (**b**) in patients with cervical cancer treated with definitive radiotherapy (*n* = 44). Dashed lines indicate median values: 9.5 mut/Mb and 22.5% for TMB and pre-RT CD8^+^TIL density, respectively.

**Figure 2 cancers-15-01210-f002:**
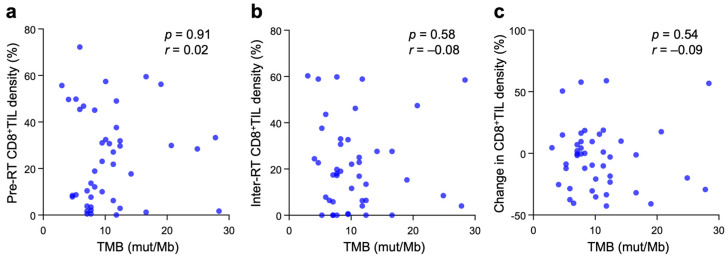
Correlation between TMB and pre-RT CD8^+^TIL density (**a**), inter-RT CD8^+^TIL density (**b**), or changes in CD8^+^TIL density between the two timepoints (**c**) in patients with cervical cancer treated with definitive radiotherapy (*n* = 44). *p* and *r* values, calculated by Spearman’s rank correlation test, are shown.

**Figure 3 cancers-15-01210-f003:**
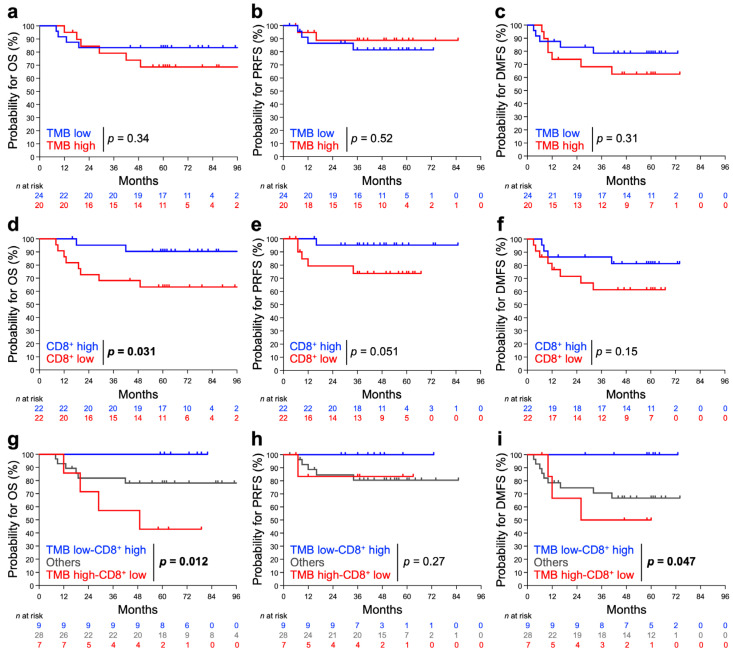
Probability of OS, PRFS, and DMFS stratified by TMB (**a**–**c**), pre-RT CD8^+^TIL density (**d**–**f**), or the two factors in combination (**g**–**i**). (**a**,**d**,**g**) OS. (**b**,**e**,**h**) PRFS. (**c**,**f**,**i**) DMSF. The patients were dichotomized with TMB or with pre-RT CD8^+^TIL density using the median: 9.5 mut/Mb and 22.5%, respectively. *p* values assessed by the log-rank test are shown.

**Figure 4 cancers-15-01210-f004:**
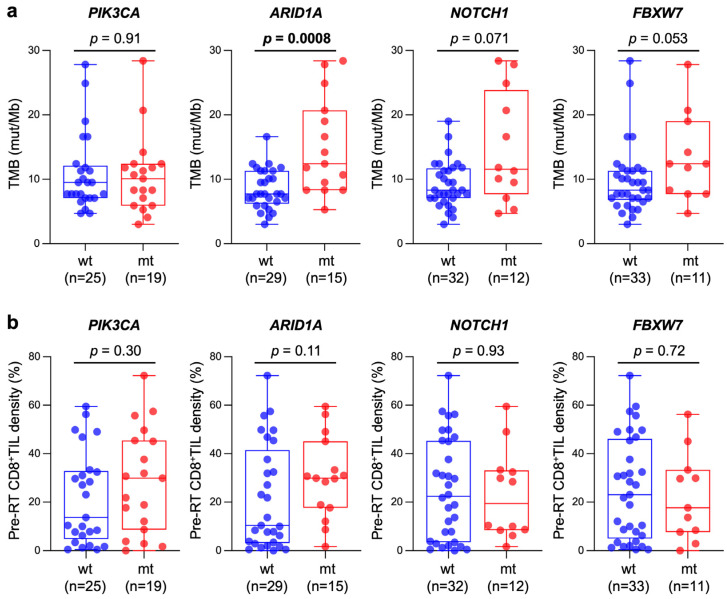
Box and whisker plots showing the association between TMB (**a**) or pre-RT CD8^+^TIL density (**b**) and mutation status of genes frequently mutated in the study cohort (i.e., prevalence >20%); wt, wild-type; mt, mutant. *p* values assessed by the Mann–Whitney U test are shown.

**Table 1 cancers-15-01210-t001:** Patient characteristics (*n* = 44).

Characteristics		Number (%)
Follow-up period (M)		61 (8–108)
Age		62 (33–87)
FIGO stage		
	IB	4 (9.1%)
	II	17 (38.6%)
	III	21 (47.7%)
	IVA	2 (4.5%)
Tumor diameter		
	≤40 mm	8 (18.2%)
	41–60 mm	24 (54.5%)
	>60 mm	12 (27.3%)
Pelvic LN status		
	Positive	24 (54.5%)
	Negative	20 (45.5%)
PALN status		
	Positive	6 (13.6%)
	Negative	38 (86.4%)
HPV status		
	Positive	34 (77.3%)
	Negative	10 (22.7%)
Concurrent CT		
	Yes	30 (68.2%)
	No	14 (31.8%)

CT, chemotherapy; LN, lymph node; M, months; PALN, para-aortic lymph node. Follow-up period and age are shown as median values (range).

**Table 2 cancers-15-01210-t002:** Association of TMB or pre-RT CD8+TIL density with patient characteristics.

Characteristics		TMB			Pre-RT CD8^+^TIL Density		
		Low	High	*p*	Low	High	*p*
		(*n* = 24)	(*n* = 20)		(*n* = 22)	(*n* = 22)	
Follow-up period (M)		61 (9–108)	62 (8–105)	0.78	61 (8–108)	62 (18–104)	0.57
Age		58 (33–87)	63 (37–80)	0.31	65 (35–87)	59 (33–77)	0.37
FIGO stage							
	IB	1 (4.2%)	3 (15%)	0.45	3 (13.6%)	1 (6.0%)	0.37
	II	10 (41.7%)	7 (35%)		8 (36.4%)	9 (40.9%)	
	III	11 (45.8%)	10 (50%)		9 (40.9%)	12 (54.5%)	
	IVA	2 (8.3%)	0 (0%)		2 (9.1%)	0 (0.0%)	
Tumor diameter							
	≤40 mm	5 (20.8%)	3 (15.0%)	0.47	5 (22.7%)	3 (13.6%)	0.18
	41–60 mm	11 (45.9%)	13 (65.0%)		9 (40.9%)	15 (68.2%)	
	>60 mm	8 (33.3%)	4 (20.0%)		8 (36.4%)	4 (18.2%)	
Pelvic LN status							
	Positive	11 (45.8%)	13 (65.0%)		11 (50.0%)	13 (59.1%)	0.56
	Negative	13 (54.2%)	7 (35.0%)	0.24	11 (50.0%)	9 (40.9%)	
PALN status							
	Positive	4 (16.7%)	2 (10.0%)		4 (18.2%)	2 (9.1%)	0.66
	Negative	20 (83.3%)	18 (90.0%)	0.67	18 (81.8%)	20 (90.9%)	
HPV status							
	Positive	18 (75.0%)	16 (80.0%)	0.73	19 (86.4%)	15 (68.2%)	0.17
	Negative	6 (25.0%)	4 (20.0%)		3 (13.6%)	7 (31.8%)	
Concurrent CT							
	Yes	16 (66.7%)	14 (70.0%)	>0.99	13 (59.1%)	17 (77.3%)	0.33
	No	8 (33.3%)	6 (30.0%)		9 (40.9%)	5 (22.7%)	

CT, chemotherapy; LN, lymph node; M, months; PALN, para-aortic lymph node. Follow-up period and age are shown as median values (range). *p* values assessed by Mann–Whitney U test (follow-up period and age) and those by Fisher’s exact test (other variables) are shown. Dichotomization with respect to TMB and that with pre-RT CD8^+^TIL density was performed using the median.

## Data Availability

The data are not publicly available due to the study protocol approved by the Institutional Ethical Review Committee of Gunma University Hospital.

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
