# Peer review of "Association between Tumor Mutational Burden, Stromal CD8+ Tumor-Infiltrating Lymphocytes, and Clinical Factors in Cervical Cancers Treated with Radiotherapy"

_cancers, 2023, doi:10.3390/cancers15041210_

Round 1
Reviewer 1 Report
Very interesting articla.
Author Response
Reviewer 1:
Very interesting article.
Response:
We sincerely thank the reviewer for evaluating our manuscript.

Reviewer 2 Report
The paper is related to very actual topic because predictive markers has yet to be deeply investigated for most cancers including cervical carcinoma. CD8+TIL density is supposed to be one of the most attractive tools for cervical cancer prognosis prediction as well as for target and radiotherapy efficacy prediction. The authors’ results concerning the comparison of TMB and CD8+TIL density is very significant and novel. Although I suppose it would be more appropriate to compare CD8+TIL density not only with TMB but with other TILs parameters and others markers which are routinely used in the labs (e.g. Ki-67 index). Because it is very important not only fundamentally understand the mechanism of their action but also to be able to implement them into routine practice. In addition, some authors demonstrated that a number of the TILs parameters can be even more effective then PD-L1 expression level for prognosis and target therapy effectiveness prediction (Mol Ther Oncolytics. 2021 Jul 21;22:410-430; Front Oncol. 2021 Jun 4;11:678758).
Author Response
Reviewer 2:
The paper is related to very actual topic because predictive markers has yet to be deeply investigated for most cancers including cervical carcinoma. CD8+TIL density is supposed to be one of the most attractive tools for cervical cancer prognosis prediction as well as for target and radiotherapy efficacy prediction. The authors’ results concerning the comparison of TMB and CD8+TIL density is very significant and novel. Although I suppose it would be more appropriate to compare CD8+TIL density not only with TMB but with other TILs parameters and others markers which are routinely used in the labs (e.g. Ki-67 index). Because it is very important not only fundamentally understand the mechanism of their action but also to be able to implement them into routine practice. In addition, some authors demonstrated that a number of the TILs parameters can be even more effective then PD-L1 expression level for prognosis and target therapy effectiveness prediction (Mol Ther Oncolytics. 2021 Jul 21;22:410-430; Front Oncol. 2021 Jun 4;11:678758).
Response:
We sincerely thank the reviewer for evaluating our manuscript and for providing insightful comments. Unfortunately, we were not able to perform immunohistochemical analysis for the molecules other than CD8+TILs due to insufficient tumor specimens. Nevertheless, we strongly agree with the reviewer's notion, and thus, this point was described in lines 334–348 with referencing the proposed two papers.

Reviewer 3 Report
I read with great interest the Manuscript titled "Association between Tumor Mutational Burden, Stromal CD8+ Tumor-Infiltrating Lymphocytes, and Clinical Factors in Cervical Cancers Treated with Radiotherapy" which falls within the aim of the Journal.
In my honest opinion, the topic is interesting enough to attract the readers’ attention. Methodology is accurate and conclusions are supported by the data analysis. Nevertheless, authors should clarify some point and improve the discussion citing relevant and novel key articles about the topic.
- The whole text should be corrected by a native English speaker in order to make the work clearer and more readable. Typos errors should be corrected.
-I find interesting a reference to the efforts made for the prevention and early diagnosis of gynecological cancers (see PMID: 36141217).
-Inclusion/exclusion criteria should be better clarified by extending their description.
- The authors have not adequately highlighted the strengths and limitations of their study. I suggest better specifying these points.
- Discussions can be expanded and improved by citing relevant articles (I suggest authors to read and insert in references the following article PMID: 33050484).
Considered all these points, I think it could be of interest for the readers and, in my opinion, it deserves the priority to be published after minor revisions.
Author Response
Reviewer 3:
I read with great interest the Manuscript titled "Association between Tumor Mutational Burden, Stromal CD8+ Tumor-Infiltrating Lymphocytes, and Clinical Factors in Cervical Cancers Treated with Radiotherapy" which falls within the aim of the Journal. In my honest opinion, the topic is interesting enough to attract the readers’ attention. Methodology is accurate and conclusions are supported by the data analysis. Nevertheless, authors should clarify some point and improve the discussion citing relevant and novel key articles about the topic.
Response:
We sincerely thank the reviewer for evaluating our manuscript and for providing insightful suggestions. According to the suggestions, we made a thorough revision on our manuscript as follows.
- The whole text should be corrected by a native English speaker in order to make the work clearer and more readable. Typos errors should be corrected.
Response:
We sincerely thank the reviewer for the important comments and apologize for the grammatical and typological errors present in the manuscript. Pertaining to this issue, Editor Dr. Jonny Yang has promised us that the Cancers editorial team will kindly send this paper to a professional English editing service after this revision round is completed.
-I find interesting a reference to the efforts made for the prevention and early diagnosis of gynecological cancers (see PMID: 36141217).
Response:
We sincerely thank the reviewer for the important comments. According to the suggestion, the explanation on the prevention and early diagnosis of cervical cancer was added in lines 42–49 with referencing the proposed paper.
-Inclusion/exclusion criteria should be better clarified by extending their description.
Response:
We sincerely thank the reviewer for the important comments. According to the suggestion, inclusion and exclusion criteria were better clarified in lines 88, 91, 92, 95, and 96.
- The authors have not adequately highlighted the strengths and limitations of their study. I suggest better specifying these points.
Response:
We sincerely thank the reviewer for the important comments. According to the reviewer's suggestion, the strengths and limitations of this study were more clarified in lines 320–348.
- Discussions can be expanded and improved by citing relevant articles (I suggest authors to read and insert in references the following article PMID: 33050484).
Response:
We sincerely thank the reviewer for the insightful suggestion. According to the suggestion, Discussion was expanded (lines 312–315) with referencing the proposed paper.
Considered all these points, I think it could be of interest for the readers and, in my opinion, it deserves the priority to be published after minor revisions.
Response:
We sincerely thank the reviewer for the encouraging comments.

Reviewer 4 Report
The authors presented a work to evaluate the association between TMB and CD8+TILs in patient with cervical cancers treated with radiotherapy. I read with great interest the manuscript, which falls within the aim of this Journal and offers a high-quality overview of the topic. The authors have performed a deep and methodologically rigorous work and offered clear and balanced conclusion to the readers, addressing future research priorities. Although the manuscript can be considered already of high quality, I would suggest to take into account the following minor recommendations:
- I suggest another round of language revision, in order to correct few typos and improve readability.
-The introduction should be extended and completed. I find interesting a reference to the combination therapy in cervical cancer (see: PMID: 35742340).
-What is the authors' point of view on the use of Parpi for the treatment of cervical cancer? (the authors can take inspiration from: PMID: 31912897).
Author Response
Reviewer 4:
The authors presented a work to evaluate the association between TMB and CD8+TILs in patient with cervical cancers treated with radiotherapy. I read with great interest the manuscript, which falls within the aim of this Journal and offers a high-quality overview of the topic. The authors have performed a deep and methodologically rigorous work and offered clear and balanced conclusion to the readers, addressing future research priorities. Although the manuscript can be considered already of high quality, I would suggest to take into account the following minor recommendations:
Response:
We sincerely thank the reviewer for evaluating our manuscript and for providing insightful suggestions. According to the suggestions, we made a thorough revision on our manuscript as follows.
- I suggest another round of language revision, in order to correct few typos and improve readability.
Response:
We sincerely thank the reviewer for the important comments and apologize for the grammatical and typological errors present in the manuscript. Pertaining to this issue, Editor Dr. Jonny Yang has promised us that the Cancers editorial team will kindly send this paper to a professional English editing service after this revision round is completed.
-The introduction should be extended and completed. I find interesting a reference to the combination therapy in cervical cancer (see: PMID: 35742340).
Response:
We sincerely thank the reviewer for the important comments. According to the reviewer's suggestion, Introduction was revised (lines 53–59) with referencing the proposed paper.
-What is the authors' point of view on the use of Parpi for the treatment of cervical cancer? (the authors can take inspiration from: PMID: 31912897).
Response:
We sincerely thank the reviewer for the insightful comments. We think that the combination use of PARP inhibitors with radiotherapy has a potential for improved outcome, considering the fact that radiotherapy induces DNA double-strand breaks which are in part repaired by homologous recombination, a synthetic lethality partner with PARP. This was described in Discussion section (lines 315–318) with referencing the proposed paper.
